# Alkyne–Alkene [2 + 2] cycloaddition based on visible light photocatalysis

Sujin Ha [1], Yeji Lee [1], Yoonna Kwak [1], Akash Mishra [1], Eunsoo Yu [1], Bokyeong Ryou [1] & Cheol-Min Park [1✉]

UV-activated alkyne–alkene [2 + 2] cycloaddition has served as an important tool to access cyclobutenes. Although broadly adopted, the limitations with UV light as an energy source prompted us to explore an alternative method. Here we report alkyne–alkene [2 + 2] cycloaddition based on visible light photocatalysis allowing the synthesis of diverse cyclo-butenes and 1,3-dienes via inter- and intramolecular reactions. Extensive mechanistic studies suggest that the localized spin densities at $sp^2$ carbons of alkenes account for the productive sensitization of alkenes despite their similar triplet levels of alkenes and alkynes. Moreover, the efficient formation of 1,3-dienes via tandem triplet activation of the resulting cyclobutenes is observed when intramolecular enyne cycloaddition is performed, which may serve as a complementary means to the Ru(II)-catalyzed enyne metathesis. In addition, the utility of the [2 + 2] cycloaddition has been demonstrated by several synthetic transformations including synthesis of various extended π-systems.

[1] Department of Chemistry, UNIST (Ulsan National Institute of Science & Technology), Ulsan 44919, Korea. ✉email: cmpark@unist.ac.kr

The synthesis of cyclobutenes has drawn much attention from the synthetic community owing to their versatility as synthetic intermediates and their presence in complex natural products[1]. Since thermal [2 + 2] cycloaddition of alkynes with alkenes is a thermally forbidden process, the synthesis of cyclobutenes has been developed primarily based on direct excitation by UV light (Fig. 1a)[2–6]. Recently, a chiral UV sensitizer has been reported for enantioselective synthesis of cyclobutenes[7]. On the other hand, various alternative methods including Lewis acid-[8–11] and transition metal-[12–16] catalyzed syntheses of cyclobutenes have been developed. However, the requirement of specific functional groups on the substrates for activation remains as limitation.

The past decade has seen a surge of developments in visible-light photocatalysis[17–20]. A number of elegant syntheses have been reported based on electron transfer (ET) photoredox processes[21–26]. Meanwhile, energy transfer (EnT) processes have drawn attention as an alternative visible-light photocatalysis in an increasing number of transformations[27–29]. For example, synthesis of N-heterocycles via sensitization of azido compounds[30–35] and isomerization of alkenes[36–38] have been described. Also, alkene–alkene [2 + 2] cycloaddition for the synthesis of cyclobutanes in inter- and intramolecular fashion based on the EnT process has been reported[39–46]. Meanwhile, only a limited number of studies on the alkyne–alkene reactions with visible light photocatalysis have been reported. Very recently, Glorius[47]

and Maestri[48] groups described novel alkyne–alkene reactions under visible light photocatalysis.

1,3-Dienes are a valuable synthetic moiety that are found in a wide variety of transformations. Intense research efforts have been made to develop efficient synthesis. For example, a number of elegant synthesis 1,3-dienes based on transition metal-catalysis including gold[49–51], palladium[52,53], and platinum[54,55]. Also, enyne metathesis with Grubbs catalysts has proven to be an efficient method for the synthesis of various 1,3-dienes[56,57]. Meanwhile, direct access to 1,3-dienes from enynes based on visible-light photocatalysis would offer a complementary route to enyne metathesis.

Here we report the visible light EnT-based alkyne–alkene [2 + 2] cycloaddition, which displays an intriguing dichotomy in reactivity with respect to the types of substrates (Fig. 1b). Whereas the intermolecular reaction affords cyclobutenes, the formation of 1,3-dienes in the case of intramolecular reaction is remarkable.

## Results

**Reaction optimization.** We began the screening with various photocatalysts under visible-light irradiation by employing di(p-tolyl)acetylene **1a** and N-methylmaleimide **2a** as the coupling partners (see Supplementary Table 1). It turned out that the use of 2.5 mol% Ir[dF(CF$_3$)ppy]$_2$(dtbbpy)PF$_6$ (**PC I**) was optimal to afford **3aa** in 83% yield. Moreover, the cycloaddition proceeded with higher yields in nonpolar solvents under diluted concentration (CH$_2$Cl$_2$ in 0.05 M). To confirm that the reaction is driven by photocatalysis, control reactions were performed, in which no reaction was observed in the absence of light or a catalyst.

At this juncture, we were prompted to investigate the underlying factors governing the reactivity of the catalysts observed during the screening (Table 1). To distinguish the two plausible reaction pathways, ET and EnT, the reduction potential ($E_{p/2}^{red} = -1.16$ V vs. SCE) and triplet energy (55.9 kcal/mol) of N-methylmaleimide **2a** were compared with those of the catalysts. A clear correlation was observed between the yields of the cycloadduct and the triplet energies of the catalysts, while their reduction potentials are inconsistent with the conversions of the reaction. For example, **PC I**, which has the highest triplet energy (60.8 kcal/mol), turned out to be the most efficient for cycloaddition, however, its reduction potential ($E_{1/2}$ (M*/M$^+$) = $-0.89$ V vs. SCE) is insufficient for the reduction of **2a**. On the other hand, a trace conversion was observed with the catalyst **PC VI**, which has a slightly higher reduction potential ($E_{1/2}$ (M*/M$^+$) = $-0.96$ V vs. SCE) but much lower triplet energy (49.2 kcal/mol) than that of **2a**. Likewise, the same trend was observed for alkyne **1a** ($E_{p/2}^{red} = -2.5$ V, $E_{p/2}^{ox} = +1.59$ V vs. SCE and $E_T = 56.7$ kcal/mol), in which the yields showed good correlation with the triplet energies rather than the redox potentials of the catalysts. These observations led us to propose that EnT process is in operation for the alkyne–alkene [2 + 2] cycloaddition, although it remains unclear which counterpart between alkenes and alkynes undergoes productive triplet excitation.

**Substrate scope.** With the optimized conditions in hand, we investigated the scope of the intermolecular reaction. First, we examined the steric and electronic influences of substituted diarylalkynes by reacting with N-methylmaleimide **2a**. It was found that both electron-rich and deficient alkynes were well tolerated (Fig. 2, **3aa–3ha**). The utility of heterocycles in bioactive compounds prompted us to examine pyridine- and pyrazine-substituted alkynes[58]. We were gratified to find that these heterocyclic substrates reacted smoothly to afford the corresponding

### a   Previous works

Cyclobutenes via [2+2] cycloaddition

Photochemical reaction (UV light)

Lewis acid (LA)-catalyzed reaction

Transition metal (TM)-catalyzed reaction

Cyclobutanes via visible light photocatalysis

Electron transfer

Energy transfer

Ring-closing enyne metathesis

Ru-catalyzed reaction
(limited substitution on enynes)

### b   This work

Visible light photocatalysis based on energy transfer process

- Cyclobutenes via [2+2] cyloaddition

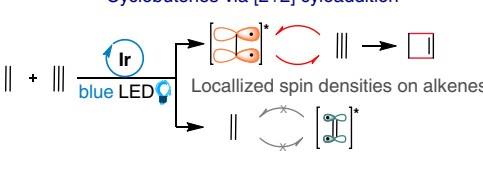

Localized spin densities on alkenes

- Formal enyne metathesis (highly substituted enynes)
- Tandem triplet activation

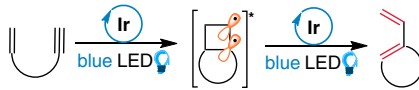

**Fig. 1 Synthesis of 4-membered ring carbocycles. a** Previous works for the synthesis of cyclobutenes, cyclobutanes, and 1,3-dienes. **b** This work for the synthesis of cyclobutenes and 1,3-dienes via visible light EnT photocatalysis.

**Table 1 Triplet energies and redox potentials of photocatalysts[a].**

| Entry | Photocatalyst | $E_{1/2}(M^*/M^+)/E_{1/2}(M^*/M^-)$ (V vs SCE) | $E_T$ (kcal/mol) | Yield[b] |
|---|---|---|---|---|
| 1[c] | PC I | −0.89/+1.21 | 60.8 | 83% |
| 2[d] | PC I | −0.89/+1.21 | 60.8 | 75% |
| 3 | PC II | −1.23/+1.40 | 60.5 | 74% |
| 4 | PC III | −0.97/+0.97 | 60.4 | 74% |
| 5 | PC IV | −0.93/+1.14 | 55.4 | 62% |
| 6 | PC V | −1.04/+1.07 | 53.0 | 35% |
| 7 | PC VI | −0.96/+0.66 | 49.2 | trace |
| 8 | Ru(bpy)$_3$(PF$_6$)$_2$ | −0.81/+0.77 | 46.5 | n.r. |
| 9[e] | Eosin Y | −1.11/+0.83 | 43.6 | n.r. |

[a]Reactions were performed with 0.05 mmol scale under Ar. Racemate for **3aa**. *p*-Tol = *p*-tolyl, bpy = 2,2′-bipyridine, n.r. = no reaction.
[b]Yields determined by $^1$H NMR spectroscopic analysis against an internal standard. (1,1,2-trichloroethene).
[c]Reaction was performed with 1.5 equiv. of **2a** and 2.5 mol% of **PC I** in CH$_2$Cl$_2$ (0.05 M).
[d]Reaction was performed with 1.5 equiv. of **2a** and 1.0 mol% of **PC I** in CH$_2$Cl$_2$ (0.05 M).
[e]Green LED instead of blue LED.

cyclobutenes in good yields (**3ia**–**3ka**). Furthermore, the reaction of alkyne **1l** bearing a cyclopropyl group gave **3la** in 67% yield with the cyclopropyl ring intact.

In addition to the aryl substitution, alkynes substituted with alkyl groups were also examined. We were pleased to find that the reaction with dialkylalkynes **1m** and **1n** proceeded to afford the corresponding cyclobutenes **3ma** and **3na**. Moreover, both silyl-substituted and terminal alkynes **1o**–**1u** smoothly participated in the reaction. We were also intrigued whether the reaction would tolerate substrates with free hydroxyl and carboxylic acid groups, which would allow obviating protecting group chemistry. Indeed, the alkynes **1s** and **1u** gave cyclobutenes **3si** and **3ua** in 85 and 78% yield, respectively. 1,3-Diyne **1v** turned out to be a good substrate to afford the corresponding alkyne-substituted cyclobutene **3va** in 70% yield.

Next, we investigated the reactivity of the alkene counterpart, and found that alkenes flanked by electron-withdrawing groups are required for efficient conversion. Thus, those flanked by the functionalities including anhydride, amide, ester, and nitrile participated smoothly in the reaction (Fig. 2, **3ab**–**3cg**). To examine the detailed electronic and steric impact of maleimide on the cycloaddition, various N-substituted maleimides were subjected to the standard conditions. It turned out that both N-H, *N*-alkyl maleimides **2h** and **2i** gave the corresponding cyclobutenes **3ah** and **3ai** in excellent yields. Various N-substituted maleimides including *N*-benzyloxy, 4-cyanophenyl, and 4-carbomethoxyphenyl malei-mides were tolerated to afford cyclobutenes in good to moderate

yields (**3aj**–**3al**). In addition, the introduction of N-heteroarenes in the cycloadduct was achieved with the maleimide **2n**. The effect of the substitution on the olefinic moiety of maleimide was further investigated. The maleimides **2o**–**2r** bearing bromo, methoxy, and methyl substituents gave the cyclobutenes in excellent yields. Dimethyl substituted maleimide **2s**, however, afforded the corresponding cycloadduct in 55% yield presumably owing to the steric hindrance. Reaction of **2t** derived from ʟ-alanine successfully proceeded to give **3at** in 93% yield.

Whereas acyclic alkenes of mono-activation such as cinnamate failed to give the corresponding cyclobutene (**3au**, see Supplementary Fig. 1a for unreactive alkenes), cyclic mono-activated alkenes including lactam **3av** and lactones (**3aw**–**3xw**) afforded the corresponding products in good to moderate yields. Moreover, a highly sterically hindered product bearing a quaternary center such as **3ax** was produced in high yield (76%). We speculate that the failure of the acyclic alkenes may be attributed to the catalyst quenching owing to the pathway involving *E/Z* isomerization[36–38]. Meanwhile, the observed reactivity of **1a** with **2w** was compared with that under direct UV-irradiation. It turned out that the reaction was sluggish under both 254 nm and 365 nm (17 and 3%, respectively; see Supplementary Fig. 1b). Also, late stage modification of commercial drugs containing alkynes proceeded smoothly when performed on *O*-acetyl 17α-ethynylestradiol and Efavirenz to afford **3ya** and **3za** in 66 and 45% yield, respectively.

A successful implementation of intramolecular alkyne–alkene cycloaddition would offer an access to valuable cyclic compounds.

**Fig. 2 Scope of the intermolecular reaction.** Unless noted otherwise, all reactions were conducted with 0.1 mmol scale under irradiation of 12 W blue LED strip and Ar atmosphere; Isolated yields; Racemates for all cyclobutenes. [a] Reaction time: 1–18 h. [b] Reaction time: 24–48 h. [c] Reaction time: 53 h–3d. [d] Isolated as N-benzylamide by in-situ treatment with benzylamine after completion of the cycloaddition; Two-step yield. [e] Reaction time: 5d; the reaction was conducted with 15 equiv. of alkene and **PC III** instead of **PC I**.

When the ester-tethered enyne **4a** was subjected to the optimized reaction conditions, coumarin **6a** was obtained in 73% yield ($E/Z = 1:1$) (Fig. 3). While the formation of diene **6a** was unexpected, we speculated that ring opening of the initial cyclobutene accounts for the diene formation. Also, the control experiments without light or photocatalyst showed no reaction (see Supplementary Fig. 2a). Thus, the photocatalyzed intramolecular enyne cycloaddition allows an access to highly substituted 1,3-dienes readily built from simple enyne substrates, which

serves as a complementary means to the enyne metathesis. Interestingly, among the ample literature precedents, the syntheses of these highly substituted 1,3-dienes based on enyne metathesis are scarce with necessitating terminal alkenes. As such, a comparison was performed by using **4a** (see Supplementary Fig. 2b). Whereas the [2 + 2] cycloaddition gave **6a** in 73% yield, enyne metathesis failed to give **6a**.

The examination of the scope of the alkyne substituents revealed broad tolerability (Fig. 3). Those substituted with aryl

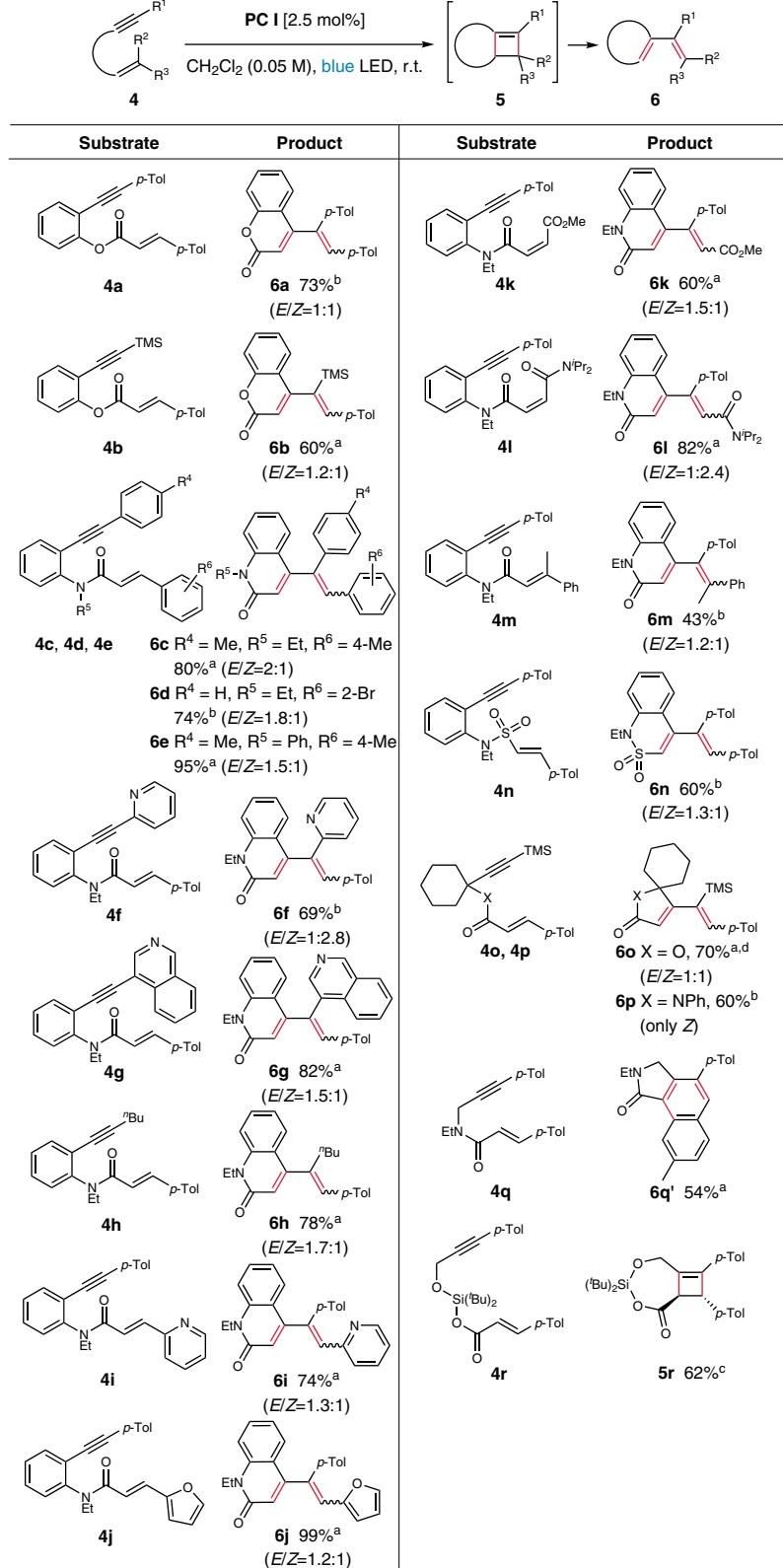

**Fig. 3 Scope of the intramolecular reaction.** Unless noted otherwise, all reactions were conducted in 0.1 mmol scale under irradiation of 12 W blue LED strip and Ar atmosphere; Isolated yields. [a] Reaction time: 1–18 h. [b] Reaction time: 24–48 h. [c] Reaction time: 60 h, racemate for **5r**. [d] Reaction was conducted with 0.05 mmol scale; Reaction concentration: 0.01 M.

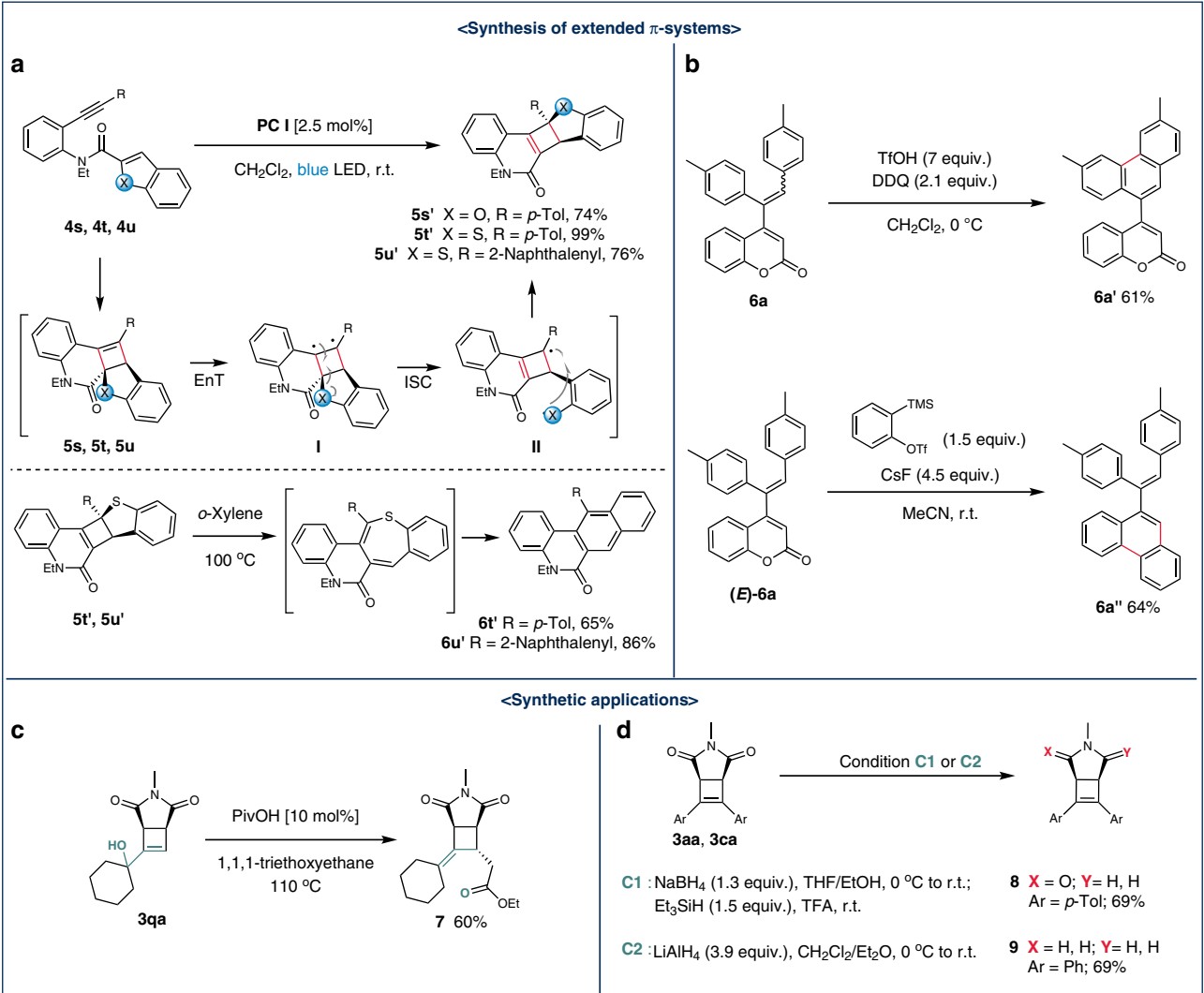

**Fig. 4 Synthesis of extended π-systems and synthetic applications.** Racemates for all cyclobutenes. **a** Tandem cycloaddition followed by rearrangement of benzofuran and benzothiophenes. (ISC = Intersystem crossing). **b** Derivatization of coumarin. (DDQ = 2,3-dichloro-5,6-dicyano-1,4-benzoquinone). **c** Synthesis of exomethylene cyclobutene. **d** Synthesis of γ-lactam and pyrrolidine derivatives.

and TMS groups gave the corresponding coumarins in good yields (**6a** and **6b**). Likewise, 2-quinolones **6c–6e** could be synthesized in good to excellent yields by employing amido tethers. Alkynes substituted with various types of substituents including aryl, heteroaryl, and alkyl groups reacted smoothly to give the corresponding 2-quinolones. For example, heteroaryl-substituted 2-quinolones **6f** and **6g** were readily prepared by employing alkynes bearing pyridine and isoquinoline groups, respectively. Also, substitution of the alkene moiety with various groups including pyridine, furan, ester, and amide was well tolerated (**6i–6l**).

The effect of steric hindrance on the alkene moiety was examined with the substrate **4m** bearing a trisubstituted alkene, which gave **6m** in albeit moderate yield. In addition to amido tethers, sulfonamide also turned out to be an effective tether providing a cyclic sulfonamide **6n**. Lastly, we examined the feasibility of 5-membered ring formation with ester and amido-tethered substrates afforded the corresponding unsaturated lactone **6o** and lactam **6p** in 70 and 60%, respectively. Interestingly, the reaction with **4q** furnished **6q′** via electro-cyclization of the corresponding diene. When silyl-tethered **4r** was subjected to the reaction conditions, cyclobutene-fused 7-membered ring product **5r** was obtained in 62% yield.

**Synthetic applications.** Extended π-systems are an important feature in various applications including fluorescence sensors and material science[59–62]. As such, we explored the accessibility to such systems based on our synthetic method (Fig. 4a, b). When we performed an intramolecular [2 + 2] cycloaddition with enyne **4s** containing benzofuran as an alkene counterpart, an unexpected product was obtained in 74%, whose structure was assigned as **5s′** (see Supplementary Figs. 346 and 347). This is in contrast to the diene formation observed in other intramolecular enynes lacking heterocyclic substituents, which arises from the ring opening of cyclobutenes. The rearrangement could be explained by that excitation of the cyclobutene intermediate **5s** leads to the formation of 1,2-diradical **I**, which undergoes fragmentation to give 1,5-diradical **II** followed by recombination to give **5s′**. This rearrangement turned out to be quite general in that the benzothiophene derivatives **4t** and **4u** afforded **5t′** and **5u′** in 99 and 76%, respectively.

To promote ring expansion, **5t′** was heated at 100 °C. Gratifyingly, tetracyclic compound **6t′** was obtained in 65% yield, which could be rationalized by the thermal electrocyclic ring opening followed by sulfur extrusion. Also, the ring expansion reaction with **5u′** proceeded well to give the rearrangement product **6u′** in 86% yield upon thermolysis. Also,

we showed that different phenanthrenes **6a′** and **6a″** could be readily prepared from **6a** by oxidative cyclization and benzyne cycloaddition, respectively.

The synthetic utility of cyclobutenes were further illustrated by the several transformations (Fig. 4c, d). Exomethylene cyclobutane **7** was prepared from allylic alcohol **3qa** by Johnson-Claisen rearrangement. In addition, 2-pyrrolidone and pyrrolidine could be synthesized in good yields by the reduction of cyclobutenes **3aa** and **3ca** bearing maleimide.

**Mechanistic studies**. To determine whether the reaction involves a radical chain mechanism, we performed light on-off experiments on both inter- and intra-molecular reactions (**3ca** and **6d**, respectively, see Supplementary Fig. 4). Conversions stopped in the absence of light in both experiments, which rules out a radical chain mechanism. This result was further confirmed by measuring the quantum yield of the cycloaddition of **1a** and **2a**. The value of 0.91 strongly supports that the cycloaddition is a non-chain reaction (see Supplementary Discussion). To further corroborate that the EnT process is operative in the cycloaddition, we performed an experiment in the presence of triplet quencher benzil ($E_T = 53.4$ kcal/mol), and found that the yield significantly decreased to 30%. (Fig. 5a).

Next, Stern–Volmer quenching experiments on several catalysts were performed by employing **1a** and **2a** as the quenchers to examine the correlation between the extent of quenching and triplet energy or redox potential (see Supplementary Figs. 7–9). The degree of quenching among the catalysts by **1a** was in good agreement with their triplet energies, not with their redox potentials, in which **PC I** with the highest triplet energy displays the most significant quenching. Likewise, the same propensity was observed with **2a**, albeit the extent of quenching was less efficient compared to **1a**. These results suggest that the cycloaddition is promoted via EnT.

An ensuing question was which counterpart between the alkyne and alkene undergoes productive triplet excitation, given their similar triplet energies (**1a** 56.7 kcal/mol vs. **2a** 55.9 kcal/mol). The analysis of the Stern–Volmer experiments indicates that **1a** is a much more efficient quencher compared to **2a**, which may suggest that alkynes excited to the triplet state react with the ground state alkenes. On the contrary, no quenching of **PC I** by alkyne **1m** was observed, which possesses much higher triplet energy

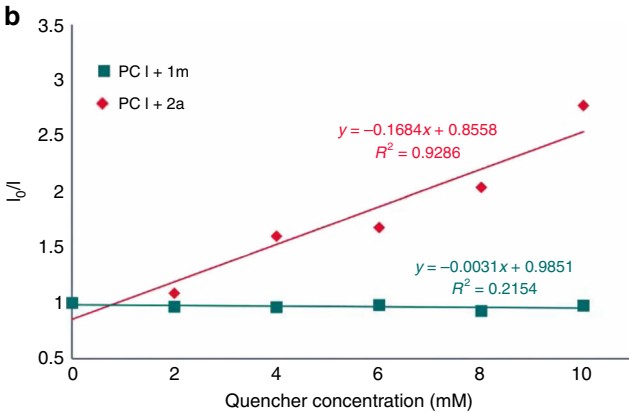

**Fig. 5 Mechanistic studies. a** Effect of triplet quencher. **b** Stern–Volmer luminescence quenching experiments using a 0.1 mM solution of **PC I** and variable concentrations of substrate **1m** and **2a** in CH₂Cl₂.

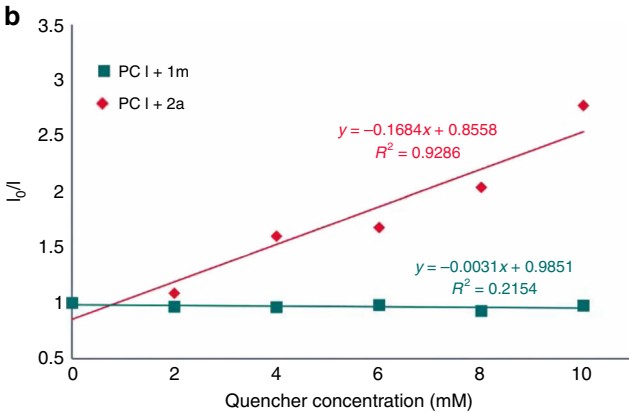

**Fig. 6 Radical clock experiments.** Racemates for all products. **a** Intermolecular reaction between cyclopropyl alkyne **1l** and **2a**. **b** Intermolecular reaction between **1a** and cyclopropyl alkene **2y**. **c** Individual reactivity of **2y** and **1l** under the standard conditions.

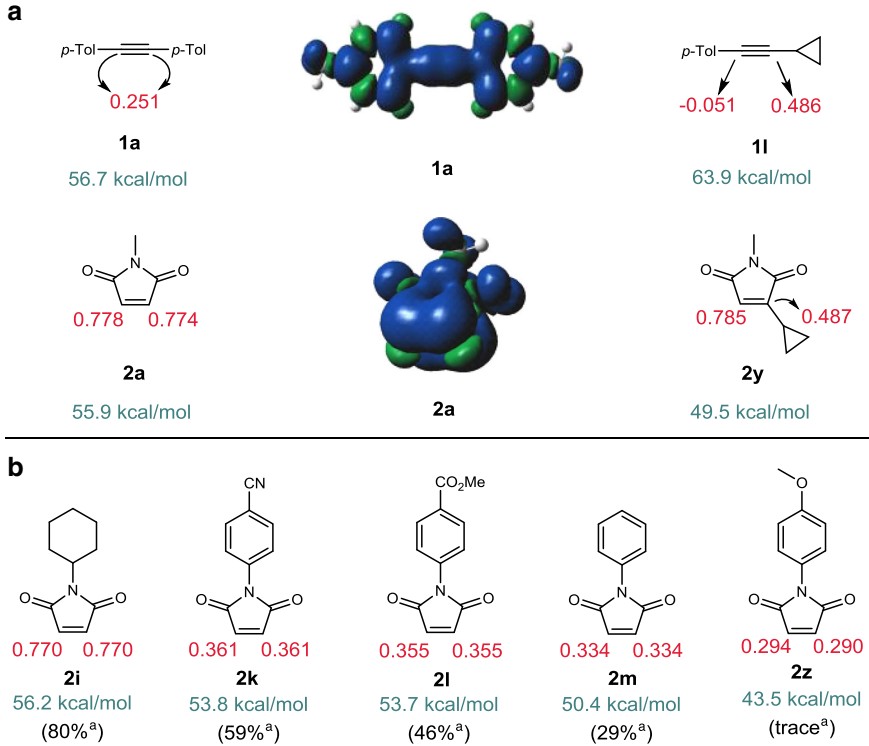

**Fig. 7 Mulliken spin densities (T₁) and triplet-singlet energy gaps of selected substrates.** [a] Reactions were conducted with the standard condition using di(p-tolyl)acetylene **1a** (0.1 mmol). **a** Spin densities and surfaces of **1a** and **2a**; Spin densities of cyclopropyl derivatives **1l** and **2y**. **b** Correlation of spin densities and reaction efficiencies of N-substituted maleimides.

($E_T = 74.1$ kcal/mol) than that of **PC I**, and yet the reaction with **2a** provides cyclobutene **3ma** in 56% (Fig. 5b), which leaves only one possibility of the participation of the triplet alkene.

To further support the rationale, we performed radical clock experiments (Fig. 6). Thus, alkyne **1l** and maleimide **2y** bearing a cyclopropyl group were prepared, and each of the substrates was reacted with **2a** and **1a**, respectively (Fig. 6a, b). Whereas the reaction of cyclopropyl alkyne **1l** with maleimide **2a** proceeded to give the corresponding cycloadduct **3la** in 67% yield, the formation of the isomerization product **2y′** along with cycloadduct **3ay** was obtained in low yields when cyclopropyl maleimide **2y** was reacted with alkyne **1a**. Furthermore, when cyclopropyl maleimide **2y** alone was subjected to the reaction conditions, **2y′** was obtained in 73% yield (Fig. 6c). On the other hand, cyclopropyl alkyne **1l** was fully recovered when subjected to the conditions. These results that **2y** undergoes ring opening upon excitation while **1l** in its triplet state remains intact could be reasoned by the lack of the radical characters at the α-cyclopropyl-substitued carbon in the triplet state **1l**. (It was reported that the rate constant for the ring opening of α-(2-phenylcyclopropyl)vinyl radicals is substantially higher than that of the corresponding α-(2-phenylcyclopropyl)carbinyl radical ($(1.6 \pm 0.2) \times 10^{10}$ s$^{-1}$ and $9.4 \times 10^{7}$ s$^{-1}$, respectively))[63,64].

To address the question, we performed DFT calculations on Mulliken spin density distributions (Fig. 7a). Whereas significantly low spin densities on the *sp* carbons of alkynes **1a** (0.251 and 0.251) and **1l** (− 0.051 and 0.486) were observed, maleimides **2a** and **2y** showed much more localized spin densities on the carbons undergoing bond formation (0.778 and 0.774, 0.785 and 0.487, respectively).

Moreover, during the survey for the scope of maleimides, we observed a wide spread in yields depending on the N-substitution. This prompted us to gauge the correlation between spin density

and reaction efficiency (Fig. 7b). A clear correlation was observed; generally, N-aryl maleimides gave lower yields compared to N-alkyl maleimides, for which the low spin density on the olefinic carbons appears to be responsible. On the other hand, maleimides substituted with electron-deficient aryl groups afforded higher yields, which is also consistent with their spin densities. Based on these experimental and computational studies, we propose that although both alkynes and alkenes undergo triplet excitation, the excited state alkenes react with the ground state alkynes to give the cycloadducts.

To examine whether the intramolecular reaction also proceeds via EnT mechanism, we performed a comparison with several photocatalysts. **PC II** and **VI** were chosen based on their triplet energies that are similar to or higher than that of **4c** (49.0 kcal/mol). As a control, those with lower triplet energies, Ru(bpy)₃(PF₆)₂ and Eosin Y, were also included. As shown in Fig. 8, the results were consistent with the triplet energies of the catalysts. Moreover, it is noteworthy that **PC VI**, which was ineffective in

**Fig. 8 Comparison of selected photocatalysts.** Reactions were performed with 0.025 mmol scale under Ar. Yields determined by ¹H NMR spectroscopic analysis against an internal standard. (1,1,2-trichloroethene). [a] Green LED was used instead of blue LED.

| PC II | PC VI | Ru(bpy)₃(PF₆)₂ | Eosin Y[a] |
|-------|-------|----------------|------------|
| 67%   | 67%   | 4%             | n.r.       |

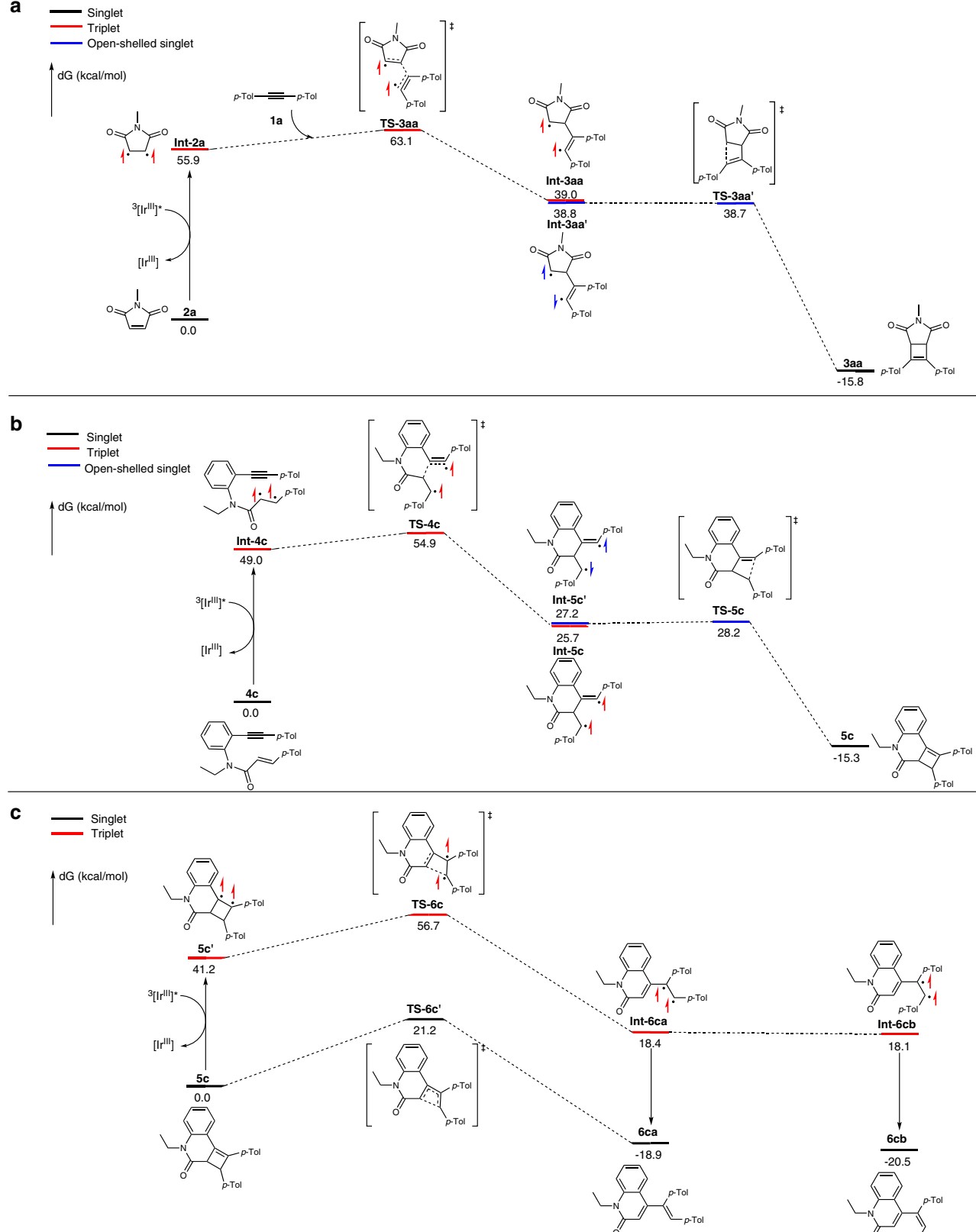

**Fig. 9 DFT calculations. a** Intermolecular cycloaddition between alkyne **1a** and maleimide **2a**. **b** Formation of putative cyclobutene intermediate **5c** from the intramolecular reaction of **4c**. **c** Formation of 1,3-dienes from **5c**.

the intermolecular reaction owing to the low triplet level relative to the maleimide (55.9 kcal/mol), provided **6c** in 67%. Also, we investigated the redox property of **4c** with cyclic voltammetry (see Supplementary Fig. 11), the low reduction potential ($E_{p/2}^{red} = -2.13, -2.44$ V vs SCE) of which makes it unlikely to undergo reduction by the catalysts examined. Likewise, oxidative pathway could be ruled out based on the oxidation potentials of the catalysts. These results indicate that EnT mechanism is responsible for the intramolecular cycloaddition.

To shed light on the reaction pathway, we performed DFT calculations on the intermolecular and intramolecular reaction pathways. All calculations were carried out with the Gaussian 09 software[65] using the M06 functional[66] with the 6-311 + g(d,p) basis set[67,68]. The SMD solvation model[69] with the solvent of dichloromethane ($\varepsilon = 8.93$) was used for all calculations. The DFT calculations on the intermolecular cycloaddition between alkyne **1a** and maleimide **2a** revealed that excitation of **2a** to its $T_1$ state (55.9 kcal/mol) by the catalyst followed by the reaction with alkyne **1a** leads to the formation of the triplet intermediate **Int-3aa** via **TS-3aa** ($\Delta G^{\ddagger} = 7.2$ kcal/mol) (Fig. 9a). Subsequently, conversion to open-shell singlet state **Int-3aa′** allows the formation of cyclobutene **3aa** via barrierless **TS-3aa′**. On the other hand, we were intrigued by the formation of 1,3-dienes instead of cyclobutenes from the intramolecular reaction. Our hypothesis was that tandem triplet activation of the initial cyclobutenes may account for the formation of 1,3-dienes. Thus, we performed DFT calculations on the reaction pathway involving the formation of 1,3-dienes via cyclobutenes as intermediates, and compared the activation barrier with that of thermal electrocyclic ring opening (Fig. 9b, c). The formation of cyclobutene **5c** is initiated by the excitation of **4c** to its triplet state (49.0 kcal/mol) by the catalyst. The addition to the alkyne to form triplet diradical **Int-5c** via **TS-4c** ($\Delta G^{\ddagger} = 5.9$ kcal/mol) followed by conversion to the open-shell singlet state **Int-5c′** via barrierless **TS-5c** results in the formation of cyclobutene **5c**.

It turns out that cyclobutene **5c** could be readily excited to its $T_1$ state (41.2 kcal/mol) by the catalyst (60.8 kcal/mol for $T_1$ state) (Fig. 9c). Rearrangement of the triplet diradical affords **Int-6ca/6cb** with the activation barrier of 15.5 kcal/mol, which results in the formation of diene **6ca/6cb**. In comparison, the activation barrier of the thermal electrocyclic ring opening via **TS-6c′** turned out to be significantly higher (21.2 kcal/mol). These results are in contrast to the cyclobutenes derived from intermolecular cycloaddition, in which a significantly higher $\Delta G^{\ddagger}$ (29.5 kcal/mol) appears to be responsible for interrupting ring opening (see Supplementary Fig. 12).

## Discussion

We developed alkyne–alkene [2 + 2] cycloaddition based on visible light EnT photocatalysis. Whereas the formation of cyclobutenes was observed from intermolecular reactions, 1,3-dienes were obtained from intramolecular reactions. For the intermolecular cycloaddition, a broad range of alkynes reacted smoothly with electron-deficient alkenes to afford the corresponding cyclobutenes. On the other hand, for the 1,3-diene formation in the intramolecular reactions, the ring opening of cyclobutene intermediates via tandem triplet excitation is responsible. Synthetically, the [2 + 2] enyne cycloaddition offers a complementary means to the Ru(II)-catalyzed enyne metathesis for the synthesis of highly substituted 1,3-dienes. Various experimental evidences support that between the two reactants, alkyne and alkene, the alkene undergoes productive excitation to a triplet state to react with the ground state alkyne. We also demonstrated the utility of the method including the synthesis of various extended π-system.

## Methods

**General procedure for the synthesis of cyclobutenes.** Alkyne (0.1 mmol, 1.0 equiv.), alkene (1.5 equiv.), and photocatalyst Ir[dF(CF$_3$)ppy]$_2$(dtbbpy)PF$_6$ (**PC I**, 2.5 mol%) were added to an oven-dried 4 mL vial equipped with a stir bar. The combined materials were dissolved in CH$_2$Cl$_2$ (2 mL) under argon atmosphere in glovebox. The reaction mixture was then irradiated by 12 W blue LED strip at room temperature (maintained with a cooling fan). After completion of the reaction as indicated by TLC, the solution was concentrated under reduced pressure. The residue was purified by flash column chromatography on silica gel to give the desired product. See Supplementary Methods for further experimental details.

## Data availability

The authors declare that all the data supporting the findings of this study are available within the paper and its Supplementary Information files, or from the corresponding author upon request.

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

## Acknowledgements

This work was supported by the National Research Foundation of Korea (NRF) grants (2014R1A5A1011165 Center for New Directions in Organic Synthesis (CNOS), NRF 2017R1A2B2012946, NRF 2017M3A9E4078558) funded by the Korean government and the UNIST Research Fund (1.170096.01).

## Author contributions

C.M.P. conceived the project. S.H. and C.-M.P. designed experiments and cowrote the paper. S.H., Y.L., Y.K., A.M., and B.R. performed experiments. E.Y. performed DFT calculations.

## Competing interests

The authors declare no competing interests.
