## [Peer Review File · Nature Communications]

Reviewers' comments:

Reviewer #1 (Remarks to the Author):

Authors reported a study on 2+2 cycloadditions promoted by visible light between an activated alkene and an internal alkyne. The reaction can be performed in an intermolecular fashion, providing an ample family of highly functionalized cyclobutenes (ca 40 examples). The intermolecular reaction of related enynes gave a complementary outcome, providing 1,3-dienes upon the formal cleavage of the alkene C=C bond. The scope and limitations of this process, too, were carefully studied (ca 20 examples). Although 2+2 cycloaddition have already been intensively studied in visible-light promoted reactions, present results are both quite original and of synthetic interest. Authors rationalized results thanks to a combined experimental and computational approach that, especially thanks to the former, is of interest for the mechanism of these sequences. Taken together, these results are highly important and therefore deserve publication in Nat Commun once the following aspects have been addressed.

Major:

Throughout the manuscript, many structures are drawn incorrectly showing the absolute rather than the relative configuration of stereocenters. Bold and dashed bonds must therefore replace their corresponding triangular version.

Minor:

The introduction presents the relevance of 1,3-dienes indicating that their synthesis is essentially limited to enyne metathesis (top of page 2); while it seems correct to claim the novelty and relevance of a photocatalytic approach to 1,3-dienes, the intro of a research article should be more complete in presenting relevant literature, in particular using gold and palladium catalysis, which are more practical routes to functionalized, cyclic 1,3-dienes of this kind compared to enyne methatesis.

Reviewer #2 (Remarks to the Author):

It is an intrinsic photophysical property of a molecule that its lowest triplet state can be populated by direct irradiation (that is via ISC from a singlet state) or by sensitization. The resulting chemistry, however, is independent from the way the triplet state has been populated. Against this background, the manuscript by Ha et al. reiterates known transformations of maleimides with alkynes for which extensive precedence exists (ref. 2, also AJC 1995, 48, 241; JOC 2005, 70, 7558; CEJ 2014, 20, 15226). The second part of the publication concerns intramolecular [2+2] photocycloaddition reactions of cinnamates which are followed by a ring opening to 1,3-dienes (photo metathesis). Here, the major criticism addresses the fact that no mechanistic proof exists for the reactions to occur via energy transfer but not via electron transfer. In general, mechanistic support for the proposed reaction pathways is weak and many hypotheses are only supported by calculations the quality of which I am not in a position to evaluate. In my opinion, the results should not be disclosed in a single manuscript but rather as a full paper on the intermolecular [2+2] photocycloaddition and as a communication on the photo metathesis. However, at this stage, I do not recommend the manuscript for publication in Nature Communications.

Reviewer #4 (Remarks to the Author):

In this manuscript, Park and co-workers report photocatalyzed intermolecular [2+2] cycloaddition of alkenes and alkynes by EnT mechanism. And they have developed photocatalytic ring close enyne metathesis for synthesis of six and five membered diene. Recently, photocatalytic [2 + 2] cycloaddition of alkenes and alkynes have been reported, but this has been less synthetic oriented and focused on mechanism studies or complex ring compound synthesis studies. The strength of

this study seems to provide information on various substrates that can be used for synthesis. And the development of enyne metathesis in intramolecular reactions is interest. In the recent Maestri and Glorius study, the mechanism of [2 + 2] cyclization through energy transfer has been reported, and the authors support the energy transfer mechanism presented by various experimental methods and computational chemistry. The results of this study show the synthetic usefulness of [2 + 2] cyclization than previous studies. Especially developing new enyne metathesis reactions is impressive. For this reason, the reviewer supports the publication of this article in Nature Communications. However, the reviewer's comments and questions must be explained before publication.

1. In the intermolecular [2 + 2] cyclization, substrates for various alkynes have been reported. However, only high active alkenes, maleimides and anhydrides etc., have reported. In particular, methyl cinnamate failed the reaction. Mono-activated alkenes is more important substrate than di-activated alkenes for use in synthesis. Thus, information on the reactivity of less activated alkenes should also be included in the paper or SI, even if failed.
2. The intramolecular enyne metathesis reaction is very interesting. What is driving force for metathesis. Ring strain or for conjugated with a carbonyl group? Explain the substrates or factors that drive [2 + 2] cyclization products and enyne metathesis products.
3. Author reported quantum yield, I recommend on/off experiment to explain proposed mechanism.

Minor points:

1. Some references (40, 48, 49, 55, 60) do not fit the form. Please correct.

Reviewer #1 (Remarks to the Author):

Authors reported a study on 2+2 cycloadditions promoted by visible light between an activated alkene and an internal alkyne. The reaction can be performed in an intermolecular fashion, providing an ample family of highly functionalized cyclobutenes (ca 40 examples). The intermolecular reaction of related enynes gave a complementary outcome, providing 1,3-dienes upon the formal cleavage of the alkene C=C bond. The scope and limitations of this process, too, were carefully studied (ca 20 examples). Although 2+2 cycloaddition have already been intensively studied in visible-light promoted reactions, present results are both quite original and of synthetic interest. Authors rationalized results thanks to a combined experimental and computational approach that, especially thanks to the former, is of interest for the mechanism of these sequences. Taken together, these results are highly important and therefore deserve publication in Nat Commun once the following aspects have been addressed.

Major:

Throughout the manuscript, many structures are drawn incorrectly showing the absolute rather than the relative configuration of stereocenters. Bold and dashed bonds must therefore replace their corresponding triangular version.

>> We greatly appreciate the reviewer's comments. Regarding the bond types, we thought that bold/dashed bonds were used, and were not sure what "triangular version" refers to. Nevertheless, we have replaced the red bold bonds with the regular bonds, and indicated that products are "racemates" as footnotes.

If we are provided with a sample, we will update as directed, or we will seek an advice from the editorial office for correction.

Minor:

The introduction presents the relevance of 1,3-dienes indicating that their synthesis is essentially limited to enyne metathesis (top of page 2); while it seems correct to claim the novelty and relevance of a photocatalytic approach to 1,3-dienes, the intro of a research article should be more complete in presenting relevant literature, in particular using gold and palladium catalysis, which are more practical routes to functionalized, cyclic 1,3-dienes of this kind compared to enyne methathesis.

>> Thank you for the suggestion. We have updated the main text with a broader view on the synthesis of 1,3-dienes.

"1,3-Dienes are a valuable synthetic moiety that are found in a wide variety of transformations. Intense research efforts have been made to develop efficient synthesis. For example, a number of elegant synthesis 1,3-dienes based on transition metal-catalysis including gold⁵⁰⁻⁵², palladium^{53,54}, and platinum^{55,56}."

Reviewer #2 (Remarks to the Author):

It is an intrinsic photophysical property of a molecule that its lowest triplet state can be populated by direct irradiation (that is via ISC from a singlet state) or by sensitization. The resulting chemistry, however, is independent from the way the triplet state has been populated. Against this background, the manuscript by Ha et al. reiterates known transformations of maleimides with alkynes for which extensive precedence exists (ref. 2, also AJC 1995, 48, 241; JOC 2005, 70, 7558; CEJ 2014, 20, 15226).

>> We greatly appreciate the reviewer's comments.

While an access to the triplet states of doubly activated alkenes based on UV irradiation are known, this manuscript describes an alternative access to triplet states based on the EnT mechanism employing visible light irradiation.

After having received the comments, we examined mono-activated alkenes for their ability to participate in the cycloaddition, and found that cyclic mono-activated alkenes including lactam **3av** and lactones (**3aw** – **3ax**) afforded the corresponding cycloadduct in good to moderate yields as shown below. Moreover, **3ax** bearing a highly sterically hindered quaternary center was produced in 76% yield.

With these results in hand, we were wondering whether UV irradiation would provide comparable results. As shown below, it turned out that the cycloaddition under both 254 and 365 nm irradiation was inefficient affording the corresponding product in 17% and 3% yields, respectively, which has been included in the main text. Therefore, these results support that the visible light-mediated EnT mechanism has its own merits.

The second part of the publication concerns intramolecular [2+2] photocycloaddition reactions of cinnamates which are followed by a ring opening to 1,3-dienes (photo metathesis). Here, the major criticism addresses the fact that no mechanistic proof exists for the reactions to occur via energy transfer but not via electron transfer. In general, mechanistic support for the proposed reaction pathways is weak and many hypotheses are only supported by calculations the quality of which I am not in a position to evaluate. In my opinion, the results should not be disclosed in a single manuscript but rather as a full paper on the intermolecular [2+2] photocycloaddition and as a communication on the photo metathesis. However, at this stage, I do not recommend the manuscript for publication in Nature Communications.

>> To address the reviewer's concerns, we performed a comparison with several catalysts shown below to probe the mechanism of the intramolecular reaction.

PC II and **VI** were chosen based on their triplet energies that are similar to or higher than that of **4c** (49.0 kcal/mol). As a control, those with lower triplet energies, Ru(bpy)₃(PF₆)₂ and Eosin Y, were included. As shown in the table, the results were consistent with the triplet energies of the catalysts. Moreover, it is noteworthy that **PC VI**, which was ineffective in the intermolecular reaction owing to the low triplet level relative to the maleimide (55.9 kcal/mol), provided **6c** in 67%.

Also, we have investigated the redox property of **4c** with cyclic voltammetry, the very low reduction potential ($E_{p/2}^{\text{red}} = -2.13, -2.44$ V vs. SCE) of which makes it unlikely to undergo reduction by the catalysts examined. Likewise, oxidative pathway could be ruled out based on the oxidation potentials of the catalysts. These results indicate that EnT mechanism is responsible for the intramolecular cycloaddition.

Photocatalyst	$E_{1/2}(M^*/M^+)$	E_T (kcal/mol)	Yield
	$E_{1/2}(M^*/M^-)$ (V)		
PC II	-1.23 / 1.40	60.5	67%
PC VI	-0.96 / 0.66	49.2	67%
Ru(bpy) ₃ (PF ₆) ₂	-0.81 / 0.77	46.5	4%
Eosin Y	-1.11 / 0.83	43.6	n.r.

$E_{p/2}^{ox} = +1.82, +0.11$ V vs SCE, $E_{p/2}^{red} = -2.13, -2.44$ V vs SCE

Reviewer #4 (Remarks to the Author):

In this manuscript, Park and co-workers report photocatalyzed intermolecular [2+2] cycloaddition of alkenes and alkynes by EnT mechanism. And they have developed photocatalytic ring close enyne metathesis for synthesis of six and five membered diene. Recently, photocatalytic [2 + 2] cycloaddition of alkenes and alkynes have been reported, but this has been less synthetic oriented and focused on mechanism studies or complex ring compound synthesis studies. The strength of this study seems to provide information on various substrates that can be used for synthesis. And the development of enyne metathesis in intramolecular reactions is interest. In the recent Maestri and Glorius study, the mechanism of [2 + 2] cyclization through energy transfer has been reported, and the authors support the energy transfer mechanism presented by various experimental methods and computational chemistry. The results of this study show the synthetic usefulness of [2 + 2] cyclization than previous studies. Especially developing new enyne metathesis reactions is impressive. For this reason, the reviewer supports the publication of this article in Nature Communications. However, the reviewer's comments and questions must be explained before publication.

1. In the intermolecular [2 + 2] cyclization, substrates for various alkynes have been reported. However, only high active alkenes, maleimides and anhydrides etc., have reported. In particular, methyl cinnamate failed the reaction. Mono-activated alkenes is more important substrate than di-activated alkenes for use in synthesis. Thus, information on the reactivity of less activated alkenes should also be included in the paper or SI, even if failed.

>> We greatly appreciate the reviewer's comments. Thanks to the reviewer's feedback, we looked at mono-activated alkenes once again and were delighted to find that the reactions with cyclic alkenes such as lactam and lactones proceeded to give the corresponding products in good to moderate yields. Also, we included the alkenes failed to give products in the SI (Supplementary Fig. 1).

2. The intramolecular enyne metathesis reaction is very interesting. What is driving force for metathesis. Ring strain or for conjugated with a carbonyl group? Explain the substrates or factors that drive [2 + 2] cyclization products and enyne metathesis products.

>> The DFT calculations show that the activation barrier for the ring opening of the initial cyclobutene intermediate **5c** from the intramolecular reaction is 15.5 kcal/mol (Fig. 9, c), which seems responsible for the facile rearrangement. We speculate that the low barrier is attributed to both conjugative stabilization and strain, as the review pointed out.

In contrast, the corresponding rearrangement of the cyclobutene **3aa** from the intermolecular reaction accompanies the activation barrier of 29.5 kcal/mol (Supplementary Fig. 12), which is twice as high relative to the intramolecular case interrupting further rearrangement.

On the other hand, to examine whether the intramolecular reaction also proceeds via EnT mechanism, we performed a comparison with several photocatalysts. **PC II** and **VI** were chosen based on their triplet energies that are similar to or higher than that of **4c** (49.0 kcal/mol). As a control, those with lower triplet energies, Ru(bpy)₃(PF₆)₂ and Eosin Y, were also included. As shown in the table, the results were consistent with the triplet energies of the catalysts. Moreover, it is noteworthy that **PC VI**, which was ineffective in the intermolecular reaction owing to the low triplet level relative to the maleimide (55.9 kcal/mol), provided **6c** in 67%.

Also, we have investigated the redox property of **4c** with cyclic voltammetry, the very low reduction potential ($E_{p/2}^{\text{red}} = -2.13, -2.44 \text{ V}$) of which makes it unlikely to undergo reduction by the catalysts examined. Likewise, oxidative pathway could be ruled out based on the oxidation potentials of the catalysts. These results indicate that EnT mechanism is responsible for the intramolecular cycloaddition.

Photocatalyst	$E_{1/2}(M^*/M^+)/$ $E_{1/2}(M^*/M^-) \text{ (V)}$	$E_T \text{ (kcal/mol)}$	Yield
PC II	-1.23 / 1.40	60.5	67%
PC VI	-0.96 / 0.66	49.2	67%
Ru(bpy) ₃ (PF ₆) ₂	-0.81 / 0.77	46.5	4%
Eosin Y	-1.11 / 0.83	43.6	n.r.

$E_{p/2}^{ox} = +1.82, +0.11$ V vs SCE, $E_{p/2}^{red} = -2.13, -2.44$ V vs SCE

3. Author reported quantum yield, I recommend on/off experiment to explain proposed mechanism.

>> As recommended, the experiment was included (Supplementary Fig. 4).

Minor points:

1. Some references (40, 48, 49, 55, 60) do not fit the form. Please correct.

>> The errors in 55 (61 in new numbering) and 60 (65) have been fixed.

However, we were not sure what were incorrect in 40 (39), 48 (47), and 49 (48). Therefore, we will seek an advice from the editorial office for correction (We thought that the standard format would require the use of "et al." where the number of authors exceeds certain limit).

REVIEWERS' COMMENTS:

Reviewer #1 (Remarks to the Author):

Authors reported a revised version of the original manuscript with present submission. The original work, for which I had already appreciated both the quality and the quantity of new results, has been significantly updated by taking into account comments from referees.

In particular, my main concern on potential misunderstanding on the drawing of products has been fixed, explicating in captions of relevant schemes that products are recovered as racemates. If authors prefer to modify drawings themselves, although not necessary as the text is clear now, I attached a chemdraw example. Other minor issues, too, were properly fixed.

Regarding comments on the generation of the putative reactive triplet, I agree with referee two that the photochemical activation of similar substrates has been already described. Present approach is therefore not conceptually new. Nonetheless, I found that the simple reaction conditions coupled with a remarkable set of synthetic applications are very important for a broad field of chemists and thus deserve publication in a generalist journal. The addition of experiments using alternative strategies to generate the desired triplet is interesting because it supports the proposed rationale for the whole sequence. Moreover it displays how the optimized method requires very finely tuned experimental conditions, balancing redox properties with triplet energies and excited states lifetimes.

Comments from referee 4 were similarly addressed. Although one might regret that some limitation apply to present method from a synthetic viewpoint, I appreciated that authors tried to further extend it through present revision and added examples of non-working substrates, too. Additional points on the mechanism and on minor bugs were similarly addressed.

Overall, I think that present submission is a significant leap forward compared to the original one, which still presented relevant findings. For these reasons, I recommend publication of present manuscript in *nat commun*.

Reviewer #4 (Remarks to the Author):

The authors have addressed the suggestions of the previous reviewers satisfactorily. All in all, I recommend publication of this work as it is.